# Acellular Small-Diameter Tissue-Engineered Vascular Grafts

**Zhen Li** [1,2,3,†], **Xinda Li** [1,2,3,†], **Tao Xu** [1,2,3,*] **and Lei Zhang** [1,2,3,*]

1 Biomanufacturing Center, Department of Mechanical Engineering, Tsinghua University, Beijing 100084, China
2 Biomanufacturing and Rapid Forming Technology Key Laboratory of Beijing, Beijing 100084, China
3 "Biomanufacturing and Engineering Living Systems" Innovation International Talents Base (111 Base), Beijing 100084, China
* Correspondence: taoxu@tsinghua.edu.cn (T.X.); stoneszhang@mail.tsinghua.edu.cn (L.Z.)
† These authors contribute equally to this work.

**Abstract:** Tissue-engineered vascular grafts (TEVGs) are considered one of the most effective means of fabricating vascular grafts. However, for small-diameter TEVGs, there are ongoing issues regarding long-term patency and limitations related to long-term in vitro culture and immune reactions. The use of acellular TEVG is a more convincing method, which can achieve in situ blood vessel regeneration and better meet clinical needs. This review focuses on the current state of acellular TEVGs based on scaffolds and gives a summary of the methodologies and in vitro/in vivo test results related to acellular TEVGs obtained in recent years. Various strategies for improving the properties of acellular TEVGs are also discussed.

**Keywords:** vascular graft; tissue engineering; acellular; small diameter; in situ

## 1. Introduction

Cardiovascular disease is one of most common diseases threatening human health and life. In China, as a result of improvements in the standard of living and ageing of the population, morbidity due to cardiovascular disease has increased, and the age of onset has decreased. Additionally, the World Health Organization estimated that 17.5 million people died worldwide from cardiovascular-related diseases in the year 2013, and this number is expected to grow to 23.6 million by 2030 [1]. Atherosclerosis is the main cause of cardiovascular disease, leading to a reduction or block of the blood flow, which results in myocardial infarction, apoplexy, as well as other severe complications. Currently, the main treatment methods for cardiovascular disease in the clinic include drug thrombolysis, percutaneous transluminal coronary angioplasty (PTCA), intravascular stents, and blood vessel bypass surgery [2].

In blood vessel bypass surgery, autologous blood vessels, such as the internal thoracic or saphenous vein, are commonly used. Repairing diseased blood vessels using autologous vessels has many advantages such as good tissue compatibility, no risk of immune reaction, and good long-term patency. However, on the other hand, autologous vessels can cause secondary trauma, and their source is limited. Therefore, artificial blood vessels are needed to take the place of autologous vessels. In addition, other clinical diseases such as congenital cardiac anomalies [3] and lower extremity arterial occlusion also create a great demand for vascular grafts.

In clinical settings, vascular grafts using synthetic materials like expanded polytetrafluoroethylene (ePTFE) and Dacron have shown good results in large-diameter blood vessel surgeries. However, for small-diameter (<6 mm) blood vessels, vascular grafts have not shown ideal results [4]. With the development of tissue engineering, the tissue-engineered vascular graft (TEVG) is a new method for

repairing blood vessels. Currently, there are various types of materials and fabrication methods used in the production of TEVGs. However, there are also some challenges related to TEVGs, such as the long in vitro preparation time and the risk of immune reactions. The use of acellular TEVGs that regenerate into autologous tissue in situ is a promising method in the future [5].

This review focuses on the current state of acellular TEVGs based on scaffolds; it provides a summary of methodologies and in vitro/in vivo test results from recent years related to this field.

## 2. Background

The aim of TEVGs is to replace autologous vessels in surgeries; therefore, TEVGs need to imitate real blood vessels as much as possible. The natural human blood vessel wall is composed of three layers: the intima layer with endothelial cells, the media layer with smooth muscle cells, and the external layer. These three layers confer the blood vessel wall good mechanical properties as well as many different functions.

TEVG fabrication methods can be categorized into two groups. One group consists of self-assembled TEVGs, and the second of scaffold-based TEVGs [6]. Scaffold-based TEVGs include three key elements [7]: the scaffold, the seed cell, and the bioreactor [8,9]. The scaffold can be prepared using natural polymers, synthetic polymers, hybrid polymers, or a decellularized matrix. Cells, such as mesenchymal stem cells [10], are used as seed cells and are seeded on the surface of the scaffold. The inner surface and the outer surface can be composed of different kinds of cells. The scaffold containing the seed cells is then placed in the bioreactor and cultured under normal conditions for weeks. During this period, cells proliferate and differentiate. As the scaffold materials (natural or synthetic) degrade, the tissue is regenerated. Finally, at this stage, the mature TEVG is obtained and is ready for further analysis.

In order to adapt to the complicated environment in vivo, the TEVG scaffold should meet the following requirements [11]:

- Mechanical properties: the TEVG must be able to maintain its shape under a certain pressure;
- Degradability: the scaffold of the TEVG should be degradable, and the rate of degradability should match the rate of tissue regeneration;
- Biocompatibility: the scaffold should not cause an immune reaction or a toxic reaction;
- Surface structure and biological activity: the inner surface of the vascular graft should be favorable to cell adhesion, growth, and proliferation.

Self-assembled TEVGs are obtained by three main fabrication methods [12]. The first is sheet-based tissue engineering. This method involves winding a 2D cell sheet around a mandrel to form a tube that is matured into a TEVG. The second method is the assembly of microtissues. Microtissues are placed in a tube-shaped mold and combined to form a TEVG. The final method is bioprinting. Cells and supporting materials are mixed and deposited in a layer-by-layer manner, and a 3D tube is constructed.

There are also certain challenges associated with TEVG fabrication, which include the requirement for a long in vitro preparation time, the difficulty of storing and transporting TEVGs, and especially, the risk of immune reaction [13]. In order to solve these problems, the acellular TEVG method was developed, which does not need an in vitro culture process and can regenerate blood vessels in situ (Figure 1). These scaffold-based TEVGs do not need long-term in vitro cultures, and the scaffold is implanted in vivo directly. After being implanted into the body, the TEVG recruits cells from the environment, and the scaffold gradually transforms into real blood vessels [14].

Current TEVGs, however, are not as suitable for vascular transplantation as autologous vessels [15]. Furthermore, there are some common complications associated with TEVGs, such as thrombosis, intimal hyperplasia, atherosclerosis, and infection, which influence the short-term and long-term patency of these TEVGs. Thrombosis is mainly caused by the absence of endothelial cells (ECs), which leads to the adherence of blood proteins on the graft lumen and the activation of clotting mechanisms. Intimal hyperplasia is mainly caused by the immigration of smooth muscle cells (SMCs). This can

have many causes, mostly associated with the mechanical properties and the implantation quality during surgery. Atherosclerosis of TEVGs occurs through a similar process to that of native vessels. Infection is mainly affected by the degree of disinfection of the synthetic scaffold.

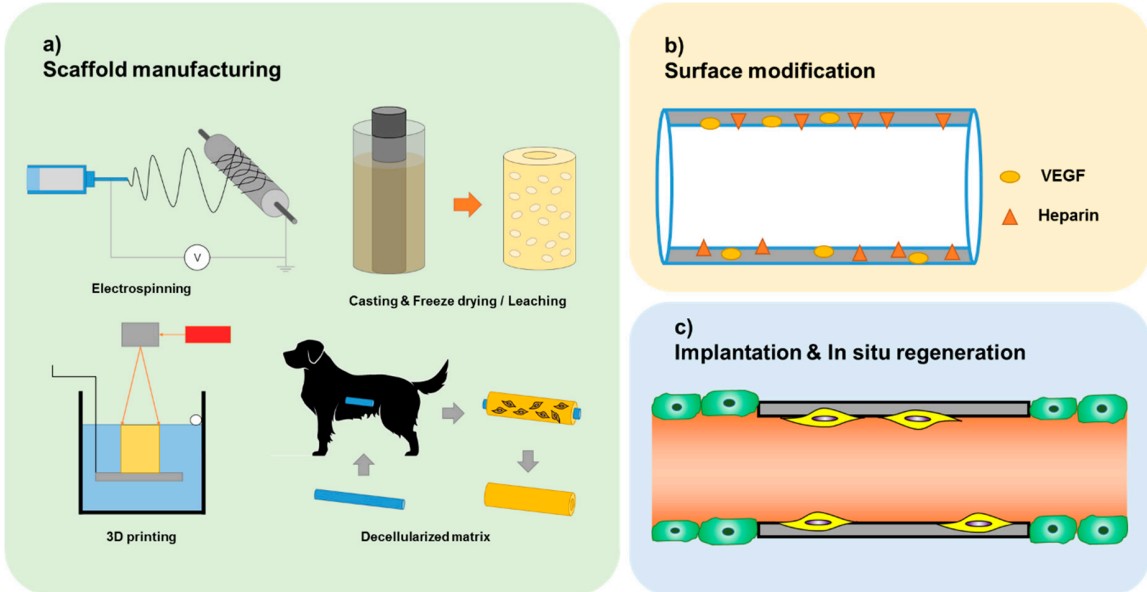

**Figure 1.** The fabrication process of acellular vascular grafts. (**a**) The scaffold of acellular tissue-engineered vascular grafts (TEVGs) can be manufactured using different methods: electrospinning, casting, three-dimensional printing (3D printing), and a technique based on a decellularized matrix; (**b**) the inner surface of the acellular TEVG can be modified with vascular endothelial growth factor (VEGF) or heparin to improve the biocompatibility and anticoagulant properties; (**c**) acellular TEVGs can be implanted directly in vivo without seed cells; the native blood vessel can regenerate in situ.

## 3. Current State of Acellular TEVG

In order to fabricate an improved acellular TEVG, various fabrication methods and modification methods are used. The scaffold of acellular TEVGs is commonly fabricated using methods such as electrospinning, casting, three-dimensional printing (3D printing), and a technique based on a decellularized matrix. Each method is reviewed in detail in the following sections.

### 3.1. Electrospinning

Electrospinning is the most common method for the fabrication of vascular grafts; this is due to its various advantages [16]. This method has a relatively low cost and a short scaffold preparation time. Various natural materials and synthetic materials can be used in the electrospinning technique, such as poly-caprolactone (PCL), polylactide (PLA), polyglycolic acid (PGA); furthermore, multiple kinds of materials can be easily combined in the same scaffold [17]. Electrospinning is an easy fabrication process owing to the simple mechanism of fiber generation from a polymer solution under high voltage.

Electrospun scaffolds have good biocompatibility because of their porous structure that mimics the natural extracellular matrix (ECM), which itself is propitious to endothelial cell attachment and proliferation. In addition, through adjusting the parameters of the fabrication process, such as solution concentration, constituent materials, voltage, jetting speed, structure of deposition platform, the various features of the electrospun scaffold can be controlled. The fiber diameter, porous ratio, thickness of the scaffold, material characteristics, as well as other features can be optimized to improve the mechanical properties and biocompatibility of the scaffold.

Electrospinning is one of the best methods for generating TEVGs cultured in vitro. However, for the in situ vascular grafts, the simple acellular scaffold cannot be implanted in vivo directly. These

electrospun scaffolds are reinforced and treated by surface modification to improve their mechanical properties and biocompatibility. All of the treatments are performed in order to collect endothelial cells in situ while avoiding the common complications associated with vascular grafts.

The mechanical properties can be optimized by combining different materials and fabrication methods. Norouzi and Shamloo [18] proposed a bilayer heparinized vascular graft that combines electrospinning and freeze-drying (Figure 2). In their study, the inner layer of the graft was made by co-electrospinning synthetic polymers, PCL, and the natural polymer gelatin (Gel). In addition, heparin was loaded by blending it in the gelatin solution and emulsion electrospinning of the PCL fibers. Adding heparin resulted in better endothelial cell attachment and proliferation, while fewer platelets were attached to the scaffold. The outer layer was fabricated using freeze-drying of a gelatin hydrogel. With an average pore diameter >200 μm, large smooth muscle cells were able to proliferate easily along this layer.

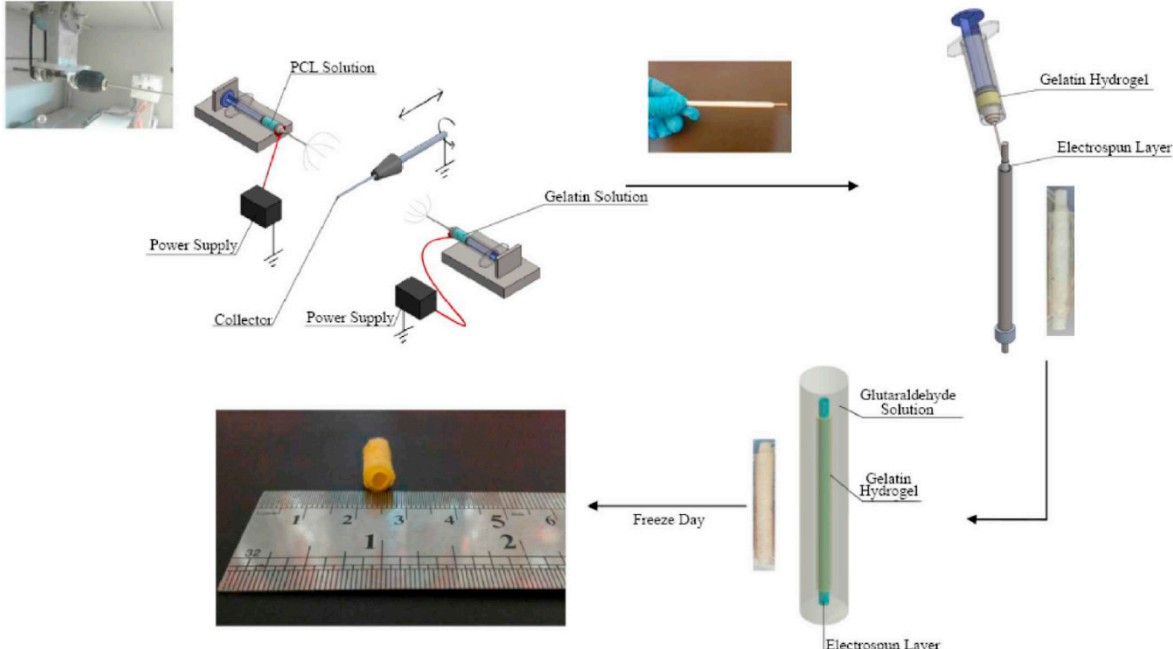

**Figure 2.** Schematic of a bilayer graft fabrication and images of a real sample at different stages. Reprinted with permission from [18], Copyright Elsevier, 2019.

Mechanical tests demonstrated the improved mechanical properties of the bilayer scaffold in comparison with the individual freeze-dried or electrospun layers.

To improve the hydrophilicity, cytocompatibility, and biodegradation of PCL vascular grafts, Shi et al. [19] fabricated a hybrid vascular graft using PCL and gelatin. This hybrid vascular graft was further functionalized by the addition of heparin, which improved the hemocompatibility. The TEVGs were implanted into rats for four weeks. The luminal surfaces were smooth at the end of the implant time. The results of the study showed that the combination of PCL and gelatin was a good method to solve some problems occurring in the TEVG related to the individual materials. Shafiq et al. [20] used a fabrication method similar to that used by Shi, mentioned above. They used heparin- and neuropeptide substance P (SP)-conjugated poly(L-lactide-co-ε-caprolactone) (PLCL) copolymers as the scaffold material because SP has been shown to accelerate tissue repair by endogenous cell recruitment and mobilization. Henry et al. [21] also adjusted the material of the scaffold to improve its mechanical properties. They blended low-molecular-weight (LMW) elastic polymers during the electrospinning process. LMW elastic polymers increased the elasticity of the scaffold and greatly increased the amount of heparin conjugated to the surface of the scaffold, which led to an increase in the load capacity of vascular endothelial growth factor (VEGF) and a prolonged the release of VEGF.

Electrospinning is the most popular method for fabricating acellular in situ vascular grafts. However, the degradation of scaffolds has not been examined in many studies. The degradation of the material has been analyzed in vitro using simulated bodily fluid, but the degradation of the vascular grafts in in vivo tests has not been analyzed. In addition, most recent case studies do not consider mechanical properties such as radial compliance, burst pressure, and suture retention strength, which are important factors for the long-time patency of a vascular graft. Furthermore, most cases only involve short-term in vivo testing for durations such as two weeks or one month, and these results do not shed light on the properties of vascular grafts in long-term use. The failure of long-term patency remains an unsolved problem for vascular grafts.

## 3.2. Casting

Casting is a traditional method for the production of TEVGs. When the graft is cast with a solution, the polymer is dissolved in a solvent. Porogen is added to the solution, and then the solution is added to a 3D mold. After the solvent evaporates, the porogen is leached, and a porous structure is formed. The pore size and the porosity can be controlled by the size and the concentration of the porogen. Compared with other methods such as electrospinning, casting is a relatively easy and quick method for fabricating scaffolds. The mechanical properties of the casting scaffold are not as good as those of other grafts; the elastic tensile modulus, suture retention force, and burst pressure are relatively low. Therefore, in order to increase the strength of the graft, different fabrication methods are combined with the casting method. The high degradation rate of the casting scaffold and the mechanical support of the outer layer fabricated by other methods make the whole scaffold suitable for vascular grafts.

Wu et al. [22] used a fast-degrading elastomer to fabricate a cell-free synthetic graft. The graft had a two-layer structure. The core layer was fabricated with poly(glycerol sebacate) (PGS) using a salt fusion and leaching method. A 1 mm mandrel and a 1.25 mm outer mold were used to form a vessel-like scaffold. To fabricate the outer layer (the PCL sheath), a PCL solution was directly electrospun onto the rotating PGS–salt template. Then, the salt was removed from the PCL–PGS–salt composites through immersion in deionized water. This study emphasized the rapid graft degradation and host remodeling, with the purpose of producing cell-free biodegradable elastomeric grafts capable of degrading rapidly to yield neoarteries nearly free of foreign materials. This design solved the problem of slow graft degradation, which limits cell penetration and causes a prolonged presence of foreign materials. Three months after interposition grafting in rat abdominal aorta, the results showed that the ECM fibers were less dense and less organized, but the majority of the graft resembled native arteries.

Melchiorri et al. [23] also combined casting methods with electrospinning, trying to obtain grafts which have the advantages of each individual method. However, the formation process was very different from that reported in the aforementioned study. First, a nonwoven copolymer bio-felt made of poly(glyocolic-co-lactic) (PLGA) approximately 0.3 mm thick was fabricated for the main structure of the scaffold. Then, the felt sections were cut into tube shapes with a diameter of 1.4 mm, and the inner lumen was formed using a 21 g stainless steel needle. A sealant solution of poly(DL-caprolactone-co-lactic acid) (P(CL-LA)) was deposited onto the scaffold to saturate the PGLA felt in order to seal and connect the seam of the rolled felt section. The test showed that this graft had good mechanical properties. After two weeks of in vivo testing in a mouse model, no evidence of thrombosis or graft aneurysm was observed. In addition, cell attachment and proliferation on the graft surface were observed, indicating adequate porosity for cellular infiltration and neotissue formation.

In order to inhibit the calcification of the graft, Sugiura et al. [24] also combined the two methods of casting and electrospinning. Two versions of bioresorbable vascular grafts were created: slow-degrading (SD) grafts and fast-degrading (FD) grafts. The inner layers of the scaffold were constructed by pouring a solution of PLCL into a glass tube and then freeze-drying under a vacuum. In order to improve the mechanical properties, these scaffolds were reinforced by electrospinning fibers on the outer surface. The outer layer of the FD grafts was composed of a combination of PLA and

PGA nanofibers, while the outer layer of the SD grafts was composed of PLA nanofibers. Both grafts were implanted in female mice for eight weeks. No bleeding, acute thrombosis, or other complications occurred in the FD groups, but several mice died from acute thrombosis in the SD group. The results demonstrated that the FD bioresorbable vascular graft was more suitable for implantation and shows promise for the prevention of graft calcification.

There are no vascular grafts which are fabricated using only casting methods due to the poor mechanical properties obtained. Electrospinning is commonly used to form the outer layer of the casting scaffold, which reinforces the whole graft. The fast degradation rate is the main reason for using this casting method. It is hoped that the scaffold material degrades quickly to facilitate cell infiltration and tissue regeneration.

### 3.3. Three-Dimensional Printing

Three-dimensional printing technology has been applied in many different fields. Bio-printing is already commonly used in tissue engineering and is also a method for fabricating TEVG [25]. As introduced previously, support materials mixed with seed cells are the starting materials for fabricating printed TEVGs. After the formation of the tube shape, this type of TEVG is further cultured in vitro. However, herein, we discuss a method for fabricating acellular scaffolds which is similar to traditional additive manufacturing techniques. Electrospinning is commonly used in TEVG because of the high porosity obtained, which increases the surface area for cell attachment and allows cell invasion. This structure and porosity can also be realized by 3D printing.

3D printing has many advantages such as the possibility of complex structure fabrication and applications using multiple types of materials, which is favorable when producing personalized TEVGs. Today, various extrusion-, inkjet-, and photocrosslinking-based additive manufacturing (AM) techniques are being applied for the fabrication of both porous vascularized tissue constructs and self-standing vascular grafts [26]. However, the high fabrication cost is a problem, compared with other methods such as electrospinning.

Melchiorri et al. [27] developed a type of 3D-printed biodegradable polymeric vascular graft. In this study, poly(propylene fumarate) (PPF) was used to construct a functional TEVG through digital light stereolithography (DLP), due to its photocrosslinking capability and its biocompatibility. The scaffold structure could be designed to fit the specific curvature of the patient's anatomy, because images of the native vessel can be obtained by MRI or CT. These printed grafts were implanted into mice for six months. No thrombosis, graft aneurysm, or stenosis was found during the six-month implantation time for all grafts.

The 3D printing method can be combined with other fabrication methods such as electrospinning. Huang et al. [28] developed a kind of triple-layer poly(e-caprolactone) fibrous vascular graft fabricated by combined electro-hydrodynamic jet (E-jet) 3D printing and electrospinning. This kind of vascular graft consisted of three layers of different structure. The middle layer, made by electrospun fibers, was fabricated first, and then the interior layer, comprising 3D-printed highly aligned fibers, was printed directly onto the middle layer. The bilayer film was wrapped over the mandrel collector, acquiring a tube shape. Finally, the exterior structure composed of mixed fibers was fabricated by co-electrospinning with PCL and polyethylene glycol (PEG). These vascular grafts were implanted into rats for 12 weeks, after which cells grew throughout the entire structure of the grafts. This triple-layer structure is obtained using 3D printing techniques to construct the lumen of the longitudinally aligned fibers of the graft to enhance the strength of the electrospun monolayer and promote cell growth and spread.

Lee et al. [29] also combined electrospinning and 3D printing techniques to fabricate a mechanically robust, bilayered scaffold (Figure 3). The inner layer was composed of an aligned electrospinning nanofibrous scaffold, which was formed by rolling the sheet scaffold around a rod. The outer layer was directly printed onto the electrospinning scaffold. The scaffold material was coated with polydopamine (PDA), and VEGF was grafted directly onto the scaffold surface to induce a potent angiogenic activity.

The coated PDA layer was evenly deposited on the bare PCL scaffold, which enabled abundant VEGF immobilization with enhanced hydrophilicity. During in vitro and in vivo testing for four weeks in rats, VEGF-immobilized scaffolds elicited markedly enhanced vascular cell proliferation and angiogenic differentiation compared with untreated groups.

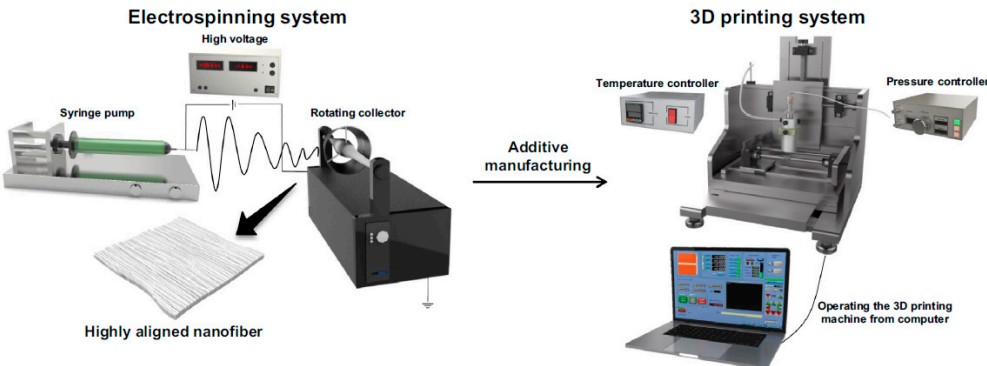

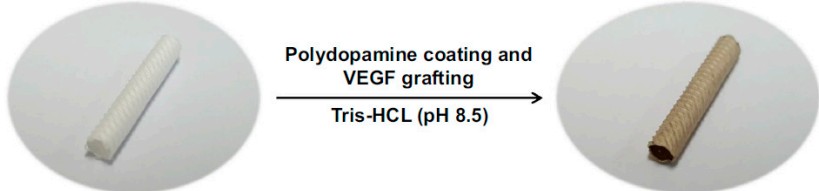

**Figure 3.** Schematic illustration of the preparation method used to generate hybrid bilayered 3D polycaprolactone (PCL) scaffolds with surface treatment for artificial vascular graft applications. Reprinted with permission from [29], Copyright Elsevier, 2019.

From our research, it is clear that 3D printing techniques are more likely to be treated as a kind of auxiliary method for fabricating vascular grafts. 3D-printed structures are dense and do not degrade easily; they usually serve as the interior or exterior layer of the multi-layered scaffold to increase the strength of the whole graft. The properties of the multilayered structures make them more suitable for long-term implantation compared with single-layer grafts such as electrospun monolayer scaffolds. The combining of materials and fabrication methods is an important trend in vascular grafting.

*3.4. Decellularized Matrix*

For TEVGs, decellularized scaffolds can be obtained from autografts, such as from the human umbilical artery [30,31] and internal mammary artery [32], or from xenografts [33]. They are seeded with cells such as stem cells and then cultured in a bioreactor. After the mature vascular graft is obtained (with good mechanical properties), an immune reaction occurs between the graft and the host. Because of this, the decellularized scaffold is developed into an acellular vascular graft, and the endothelial cells are collected in the body, while the tissue regenerates in situ. Here, we do not discuss the decellularized scaffold from real blood vessels but the in-body tissue architecture (iBTA), a kind of cell-free in vivo tissue engineering technology to produce autologous implantable tissue of the desired shape. A kind of tubular mold is implanted in the abdominal cavity or subcutaneously, using the body microenvironment as a bioreactor. Then, the cells in the body grow on the tubular mold to form a cell wrap layer, and this living tissue is further decellularized and used as a vascular graft. In addition, nonvascular tissues, such as amniotic membranes and the small intestinal submucosa (SIS), have also been decellularized and used to produce TEVGs [12].

This acellular extracellular matrix vascular scaffold is composed of a natural extracellular matrix. It has the structure and mechanical performance of natural tissue ECM, but the mechanical strength and compliance are not ideal for use as a vascular graft scaffold.

Nakayama et al. [34] developed vascular tubular tissues called biotubes, which consist of autologous tissues. Researchers embedded six kinds of polymeric rods as molds in subcutaneous pouches in the dorsal skin of New Zealand white rabbits. After months of implantation, the biotubes were constructed. Differences in the material influenced how long it took for stable biotubes to form, while also influencing the wall thickness of the biotubes. It is hoped that the biotubes can be prepared as vascular grafts in a patient's body using autologous cells and extracellular matrix components.

After a decade, Nakayama et al. [35] used iBTA to develop biotubes of a higher quality compared to those previously fabricated (Figure 4). An inner spiral mandrel and an outer pipe-shaped spiral shell were assembled to be used as spiral molds for long biotube formation. The molds were embedded into subcutaneous pouches in dogs or goats for one or two months. There were slits on the spiral shell so that autologous cells could move to the space between the mandrel and the shell to form the biotube wall (made of collagenous tissues). The biotubes could be used for implantation with no need for luminal modification or mechanical support. The result demonstrated vascular reconstruction within three months of implantation into dogs.

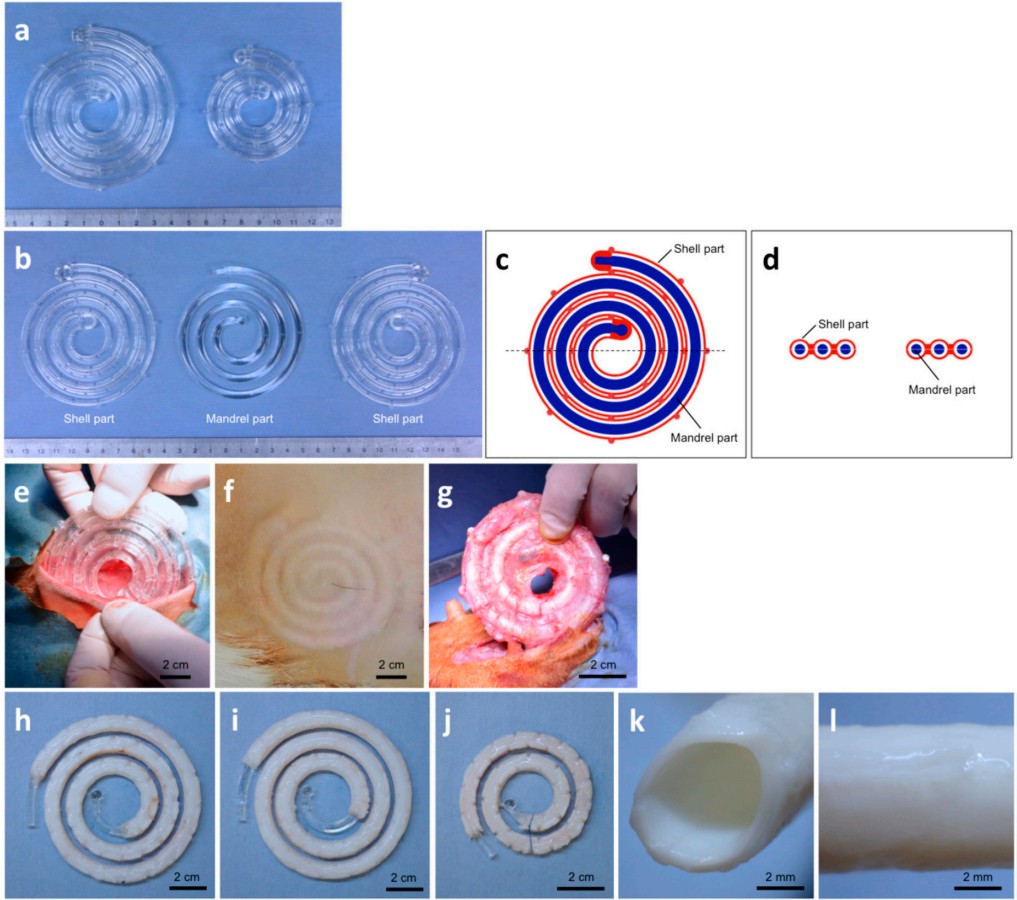

**Figure 4.** Photographs of (**a**) two spiral-shaped molds used for the preparation of long biotubes of 50 cm (left) or 25 cm (right); (**b**) three parts for the molds; (**c**) cross-sectional view of the mold in the horizontal direction, (**d**) and perpendicular direction at the dotted line in c; (**e**) subcutaneous embedding of the mold into a goat, (**f**) in vivo incubation, and (**g**) harvesting of the mold from a goat. The long biotubes formed around the spiral mandrels after embedding 50 cm spiral molds for (**h**) 1 or (**i**) 2 months or (**j**) obtained from 25 cm spiral molds after two months; (**k**) the luminal surface and (**l**) side view. Reprinted with permission from [35], Copyright Elsevier, 2018.

Smith et al. [36] developed a heparin/VEGF-functionalized cell-free vascular graft made of SIS. This kind of vascular graft was implanted in a mouse model for one month, and the results showed that the grafts maintained patency and promoted the regeneration of a functional endothelium.

For grafts which require an implantation mold in the abdominal cavity or subcutaneously, although they have advantages, such as natural tissue structure and avoidance of immunological rejection after implantation, their strength is not sufficient, and the cost and time requirements are prohibitive. All of these facts make the application of this kind of graft a significant challenge.

## 4. Discussion

Recent studies into TEVGs have been mainly focused on acellular TEVGs, which lead to in situ blood vessel regeneration. The goal of all these studies was to produce TEVGs which satisfy as much as possible the main requirements of well-matched mechanical properties, blood compatibility, endothelialization, and biodegradability [37].

Until now, all TEVGs have presented the same problem: the scaffold cannot satisfy all of the necessary mechanical properties at the same time. These include elastic (Young's) modulus, radial compliance, burst pressure, and suture retention strength. A mismatch of any single mechanical property between the graft and the native blood vessel of the host leads to low patency for long-term implantation. Low burst pressure causes the TEVG to expand or even rupture, which directly leads to the failure of the TEVG. Insufficient suture retention strength leads to leakage at the pinhole after implantation surgery. The requirements of burst pressure and suture retention strength can commonly be met by TEVG, but compliance remains a problem. Because of the complex environment in vivo and the degradation process of vascular grafts, we cannot determine which radial compliance is most suitable for implantation. Moreover, unsuitable compliance leads to a hemodynamic difference between the environment near the graft and the real environment. This situation can reduce blood flow velocity, which results in thrombosis and intimal hyperplasia at the anastomotic site [6]. To this end, many research groups are working to combine different fabrication methods and material types in order to construct a scaffold with better mechanical properties.

Blood compatibility is basically satisfied by most TEVGs through the proper selection of scaffold materials; this mainly requires low cytotoxicity and surface modifications such the addition of heparin to improve hydrophilicity and anticoagulant properties.

Rapid endothelialization is still one of the core problems related to the use of TEVGs. The lack of endothelium cells or the lack of the formation of the EC film on the inner surface of the TEVG generally leads to thrombosis. Particularly for acellular TEVGs, the whole endothelium process occurs in vivo; there are no seed cells or in vitro cultures to accelerate the endothelium process. All ECs need to be recruited from the nearby natural blood vessels. Moreover, the proliferation and attachment processes are of great importance. These concern the microstructure and further surface modifications. The high porous ratio obtained, which is beneficial for proliferation and attachment of ECs, is the main reason for choosing the electrospinning method. Surface modifications, which lead to a higher load capacity of VEGF, were used by many research groups in the attempt to improve the rate of endothelialization.

Biodegradability is a crucial factor for the scaffold. A mismatch between the scaffold material's degradation rate and the tissue regeneration rate leads to a change in the mechanical properties of TEVGs after implantation, which can cause different kinds of failures such as TEVG bursting or occlusion. Many groups use in vitro testing to simulate the degradation process, such as bioreactors [38]. In real circumstances, however, it is very difficult to calculate the rate of degradation and scaffold regeneration because of the countless disturbance factors. This unsolved problem requires further analysis and experimentation.

The final desired result is long-term patency after implantation. All the in vivo testing in animal models aims to provide more positive experimental results and favorable support for clinical experiments. However, for small-diameter vascular grafts, it is difficult to meet all the requirements at the same time. Nearly all studies include in vivo testing only for several weeks or months; long-term

(>6 months) in vivo testing has barely been reported (the results of recent in vivo testing are shown in Table 1). As for animal models, compared with small animals, the in vivo environment of larger animals is more similar to that of humans. Therefore, good results related to long-term in vivo testing in large animals are more convincing than those for small animals. However, small animals such as rats or rabbits are more commonly used than large animals like canines or sheep [39]. There is still a great deal of research to be done in order to obtain ideal results from long-term in vivo testing.

## 5. Conclusions

In conclusion, acellular TEVGs are considered an effective type of small-diameter vascular grafts that can lead to in situ regeneration. However, long-term patency is still the core problem associated with vascular grafts. Different scaffold materials, fabrication methods, and surface modification methods have been investigated with the aim of improving the mechanical properties, biocompatibilities, and degradability of TEVGs (Figure 5). Moreover, many in vivo tests have been performed in the hope of showing long-term patency. However, as a result of the complicated process of in situ regeneration, it is difficult to judge which degradation rates are most appropriate. In addition, the question of the ideal mechanical properties of the scaffold is still open to debate. More research on the subject is needed, but undoubtedly, ideal TEVGs are gradually becoming a reality.

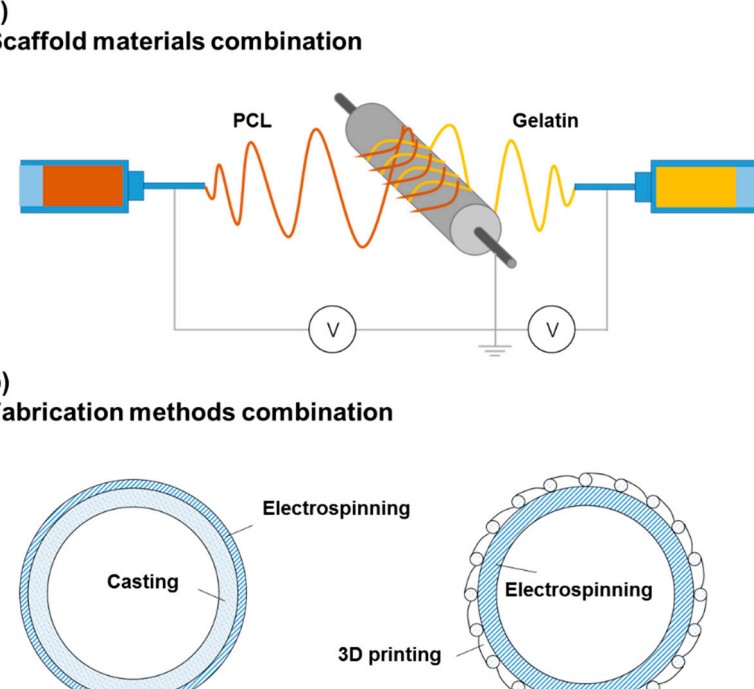

**Figure 5.** Recent progress regarding TEVGs. (**a**) The scaffold of the TEVG was fabricated by co-electrospinning, which combined different types of materials. (**b**) The multilayered structure of the scaffold was constructed by combining different types of fabrication methods.

**Table 1.** In vivo test of acellular TEVG in recent years.

| Year | Manufacturing Method and Structure of Scaffold | Animal Model | Duration (Week) | Patency | Reference |
|------|-----------------------------------------------|--------------|-----------------|---------|-----------|
| 2019 | small intestinal submucosa (SIS) functionalized with heparin/vascular endothelial growth factor (VEGF) | mice | 4 | 100% | [36] |
| 2018 | decellularized film tissue obtained by implanting a mold in vivo in dog or goat | dogs and goats | 12 | - | [35] |
| 2018 | electrospinning of poly-caprolactone (PCL) and gelatin surface modification with heparin | rats | 4 | 100% | [19] |
| 2018 | electrospinning of PCL for inner layer 3D printing of polydopamine (PDA) for outer layer immobilization of VEGF | rats | 4 | 100% | [29] |
| 2018 | electrospinning of a poly(ε-caprolactone)-b-poly(isobutyl-morpholine-2,5-dione)/silk fibroin (PCL-PIBMD/SF) mixed solution surface modification with polyethylene glycol (PEG) and cell-adhesive peptides | rabbits | 10 | 100% | [17] |
| 2018 | casting of poly(glycerol sebacate) (PGS) for inner layer electrospinning of PCL for outer layer | rats | 12 | 90.5% (19/21) | [40] |
| 2018 | wet spinning of circumferentially orientated poly(L-lactide-co-ε-caprolactone) (PLCL) for inner layer electrospinning of PLCL for outer layer | rats | 48 | - | [41] |
| 2018 | 3D printing of aligned structure PCL fiber for inner layer electrospinning of PCL for middle layer co-electrospinning of PCL and PEG for outer layer | rats | 12 | - | [42] |
| 2018 | electrospinning of PCL surface modification with heparin and neuropeptide substance P | rats | 4 | 100% | [28] |
| 2017 | electrospinning of polylactide (PLA) and polyglycolic acid (PGA) for inner layer casting of PLCL for outer layer | mice | 8 | 100% | [24] |
| 2017 | electrospinning of poly-L-lactic acid (PLLA) and PCL surface modification with heparin and VEGF | rats | 4 | 88% (10/11) | [21] |
| 2017 | electrospinning of PCL and polydioxanone (PDS) | rats | 12 | - | [43] |

**Table 1.** *Cont.*

| Year | Manufacturing Method and Structure of Scaffold | Animal Model | Duration (Week) | Patency | Reference |
|---|---|---|---|---|---|
| 2017 | electrospinning of polyurethane (PU) surface modification of stromal cell-derived factor-1$\alpha$ and VEGF | dogs | 24 | 62.5% (5/8) | [44] |
| 2017 | electospinning of polycarbonate polyurethane (PCU) surface modification of NH$_3$ plasma and heparin | rabbits | 4 | - | [45] |
| 2016 | 3D printing of poly(propylene fumarate) (PPF) | mice | 24 | 100% | [27] |
| 2016 | decellularized film tissue obtained by implanting a mold in vivo in pigs for inner layer electrospinning of PCL for outer layer | pigs | 4 | 87.5% (7/8) | [46] |
| 2013 | comprising collagen fiber networks and elastin-like protein polymers | rats | 2 | 100% | [47] |
| 2012 | electrospinning of PU | dogs | 24 | 83% (5/6) | [48] |
| 2008 | electrospinning of PCL | rats | 24 | 100% | [49] |

**Author Contributions:** T.X. and L.Z. proposed the idea; L.Z., Z.L., and X.L. conceived and designed the structure of this review; Z.L. contributed the materials, figures, and tables for this review; Z.L. wrote the paper.

**Funding:** This research was partially supported by Beijing Municipal Science and Technology Commission (Grant No. Z141100000214012).

**Conflicts of Interest:** The authors declare no conflict of interest.

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
