# Peer review of "Acellular Small-Diameter Tissue-Engineered Vascular Grafts"

_applsci, doi:10.3390/app9142864_

Reviewer 1 Report

Compared to previous review articles in similar topics, such as [37] Wu et al. 2018 or [12] Pashneh-Tala, the authors' review manuscript may not add much helpful information to readers. Although Table 1 with References Bibliography may inform the readers some recent research articles published in 2016-2019 (after the previous review articles), this reviewer could not find authors' elaborated review of those in their manuscript text, except only few articles under each subsection of section 3 and Table 1.

Authors emphasized the phrase of 'small-diameter' TEVG in the title/abstract/keywords/introduction, which seems not reviewed/mentioned in detail for each article. Authors may consider rewriting the manuscript focusing on such information in the text, to differentiate this review article compared to previous ones. Authors may also show the small-diameter information in the table 1 as well.

To this reviewer, some figures are somewhat redundant to each other and don't aid reading of text that much.

Overall structure of the manuscript writing can also be improved. Background is generally included in the Introduction. Electrospinning section was already described in 3.1., which keeps recurring as a hybrid method with casting (3.2) or 3D printing (3.3).

Some of the manuscript text were not polished (including typos, use of abbreviations without spell-out, etc), which needs further elaboration. The English used is readable, but overall/moderate English corrections is necessary to correctly deliver the authors' message to readers. Also, please double-check the wording of 'endothelium friendliness' is generally accepted term in the field. To this reviewer's knowledge, 'friendliness' is not much used in scientific articles while it is still readable.

Reviewer 2 Report

This review focuses on the current state of the new tissue vascular grafts based on scaffolds,

and the authors provides a clear summary of methodologies and in vitro/in vivo test results  to creates scaffold-based TEVGs

The paper is clear and well structured.

Author Response

Thank you very much!

Reviewer 3 Report

I find this review article relevant to the field. I do feel that the review could include more details in depth throughout each section. In the Background, four requirements were mentioned. In-depth discussion regarding those four requirements could be included in each section (electrospinning, casting, 3D printing, and decellularized matrix)

A few suggestions:

Page 1, line 45 : Can authors please define what ideal results are?

Page 2, line 93 and 95 : Please define ECs and SMCs.

Page 5, line 167 : Can authors explain what porogen is and why it is added to the solution?

Figure 3 needs copyright.

References in the text : Use the last name of the first author, instead of the first name of the author. For example, Anthony et al. [23] (page 5, line 188) should be Melchiorri et al.

In Electrospinning section (3.1): Can authors elaborate more on degradation of mentioned vascular grafts in the end?

In discussion: Mechanical properties and elastic Young’s modulus were mentioned. Can authors explain in details what ideal mechanical property/elasticity is, e.g., 1 kPa or 3-4 MPa? And how do TEVGs differ from each other and native blood vessels?

Author Response

Round  2

Reviewer 1 Report

Ref. [37] may not have been written by native English speakers. Except the ref. [37], could the authors find more scientific articles using the term of 'endothelium friendliness' ? Probably very few.

Again, please double-check whether the wording of 'endothelium friendliness' is generally accepted term in the field and revise appropriately.

The English used is readable, but overall/moderate English corrections is necessary to correctly deliver the authors' message to readers. If authors can have colleagues of native English speaker, please have the colleagues to proof-read the manuscript before submission. If not, there are English language editing services available for non-native authors (although those are not free).
